# ERRORMAP AND ERRORATLAS: CHARTING THE FAILURE LANDSCAPE OF LARGE LANGUAGE MODELS

## ABSTRACT

Large Language Models (LLM) benchmarks tell us when models fail, but not *why* they fail. A wrong answer on a reasoning dataset, for instance, may not reflect weak reasoning at all, but instead a formatting slip, a calculation error, or dataset noise. Without disentangling such causes, benchmarks give an incomplete picture and cannot reliably guide model improvement. We introduce `ErrorMap`, the first method to systematically chart the sources of LLM failure. `ErrorMap` provides tools to extract a model's unique "failure signature", uncover what benchmarks actually measure in practice, and broaden the scope of identified model errors to reduce blind spots. This enables developers to debug models more effectively and helps benchmark creators align dataset goals with actual outcomes. Additionally, it supports benchmark consumers in identifying which models best suit their specific needs. `ErrorMap` is designed to work flexibly with any model and dataset, making it adaptable to evolving architectures and emerging data sources without requiring changes to its logic. We apply our method across 21 datasets and 73 models to automatically generate `ErrorAtlas`, a taxonomy of model errors, revealing recurring failure patterns in current language models. `ErrorAtlas` highlights error types that are currently underexplored in LLM research, such as omissions of required details in the output and question misinterpretation. By shifting focus from where models succeed to why they fail, `ErrorMap` and `ErrorAtlas` lay the foundation for next-generation evaluation — one that exposes hidden weaknesses and directs meaningful progress. Unlike success, which is typically measured using task- or dataset-level metrics, our approach introduces a deeper layer of evaluation that can be applied globally across models and tasks, offering richer insights into model behavior and limitations. We make the taxonomy and method code publicly available[1], with plans to update `ErrorAtlas` as new benchmarks emerge.

*"It is possible to fail in many ways. . . while to succeed is possible only in one way."*

*Aristotle,* Nicomachean Ethics*, Book II,* ∼*320BC*

## 1 INTRODUCTION

Benchmarking plays a central role in advancing large language models (LLMs), offering a standard bottom-line score to assert progress (Biderman et al., 2024). This abstraction eases proving a model's success or its overall superiority, but it also obscures the nature and origin of model errors, complicating skill comparisons and hindering efforts to diagnose limitations or guide improvements.

Figure 1: A demonstration of some high-level `ErrorAtlas` categories, each illustrated with two label examples.

[1]https://anonymous.4open.science/r/ErrorMap-BDBC

In response to these limitations, there is growing interest in developing more interpretable and diagnostic evaluation frameworks (Maimon et al., 2025; Zeng et al., 2025; Tjuatja & Neubig, 2025) or in highlighting specific errors models make (Mukherjee et al., 2025; Pan et al., 2025; Li et al., 2024; Honovich et al., 2022; Kryscinski et al., 2019). While current diagnostic methods offer valuable insights, their analysis primarily relies on the challenges posed by the input (e.g., counting failures on differential equations questions). However, where success on a challenge necessarily proves competence, a failure can have many causes (e.g., misunderstanding the question, miscalculation or applying a wrong axiom). Moreover, benchmark examples themselves may introduce ambiguity or error, further complicating evaluation, especially when the model's answer is not considered. Ultimately, pinpointing the cause of failure requires analyzing both the input, including the question and the instruction, and the resulting answer.

We introduce `ErrorMap` (§2) to address this gap. `ErrorMap` offers a model-oriented, rather than data-oriented error analysis, highlighting why models fail. The method transforms raw language model failures into a structured, interpretable taxonomy in natural language. To do so, `ErrorMap` follows a pipeline, first profiling the issue underlying each failure, then generating high-level taxonomy categories in an iterative refinement stage and finally applying the generated taxonomy to each failure. As our analysis requires simple unstructured text, it applies seamlessly to any language model and domain. Overall, `ErrorMap` provides a flexible way to analyze a practitioner's specific setting or to compare the failure fingerprint of several models or datasets. While a dynamic taxonomy that would fit an ad hoc analysis is best for many needs, a stable taxonomy simplifies comparisons across time and replicability.

Applying `ErrorMap` to 73 models and 21 datasets, we release `ErrorAtlas` – a taxonomy of current LLM failures (see Fig. 1, §3). `ErrorAtlas` is a comprehensive and static taxonomy of model errors designed to facilitate cross-field comparisons, enhance efficiency, and ensure replicability.

We present several findings on common model failures, which both stand on their own and highlight the effectiveness of our methods. Applying the `ErrorAtlas` taxonomy to current models and datasets we find (§4) issues that are prevalent but understudied. For example, models often misinterpret the question's intent and often provide incomplete answers. We find different error patterns across model families and types, and find for example that Llama (Grattafiori et al., 2024) models of different sizes have similar error distributions, but instruct and turbo models differ radically.

Beyond `ErrorAtlas`, `ErrorMap` can support practitioners throughout the LLM lifecycle, from development and fine-tuning to evaluation and benchmarking. We provide in §5 two test cases, a model developer diagnosing the differences between two versions of Gemini (Team et al., 2024) and a benchmark curator testing MMLU-pro (Wang et al., 2024a).

Furthermore, in §6, we validate that the stages involved in extracting the error analysis are accurate, robust and cover well the errors models make.

Our main contributions:

1. We introduce `ErrorMap`, an LLM-based technique to generate a dedicated taxonomy of LLM errors. It enables analysis across a diverse set of domains, input formats and model comparisons.
2. We present `ErrorAtlas`, a static taxonomy of model errors, generated using `ErrorMap`. It captures common failure modes across benchmarks and models. These errors reflect underlying limitations in model behavior, supporting meaningful and interpretable comparisons of model weaknesses.
3. We provide analysis across a large number of models and datasets. Finding common errors that are understudied. Moreover, we find model versions, types, and families exhibit distinct error patterns, allowing for nuanced behavioral profiling and more targeted evaluation.
4. We demonstrate the applicability of both `ErrorMap` and `ErrorAtlas` for nuanced model comparison, benchmark analysis, and model debugging.
5. We publicly release the code, taxonomy and associated data.

## 2 ERRORMAP

Our technique targets a common scenario: evaluating multiple models on the same dataset examples, as typically done in benchmark runs, or even reusing a benchmark run for deeper analysis. It

leverages all available data in the benchmark, including inputs, reference answers, and model outputs, and produces comparative insights across models. The process is unsupervised and consists of three stages: (1) analyzing incorrect predictions on a per-example basis, (2) extracting and iteratively refining error categories, and (3) applying the error categories to the incorrect predictions, to organize them into a structured representation, resulting in a layered taxonomy of error types. We provide additional information including the specific prompts used in Appendix A.

**Stage 1: Per-Instance Error Analysis** Our goal at this stage is to create a structured summary of the resulted errors and provide interpretable analysis of it. To achieve this, we task an analyst LLM with performing a detailed, structured analysis for each incorrect prediction. This includes evaluating a list of criteria with associated features, providing a summary of the failure, and assigning a short *label* for it. To support the judge's evaluation, we provide the following information: the original instance, any available references and multiple Informative Correct Predictions (ICPs) if available, i.e., correct predictions made by other models in the benchmark. ICPs have proven useful (Zelikman et al., 2022; LI et al., 2022; Creswell & Shanahan, 2022). In this context, they act as rich reference points that often approximate full solutions, helping judges compare correct and incorrect outputs rather than diagnose root causes. This is particularly valuable when no gold reference exists or when the gold standard is limited to a final answer (e.g., in classification tasks).

The judge is asked to construct a structured solution to the instance (see prompt in Appendix A.1.1.) The structure has several components, all of which the judge should fill. The judge is asked to break down the solution and specify *criteria*; steps, evidence, or assumptions required to reach a correct answer, for example relying on formulas, a list of reasoning steps of extracting multiple facts to deduce and answer. For each criterion, the judge should assess its presence, quality, supporting evidence (a quote from the prediction), and may add comments if there is something additional to note about this criterion. Grounded in the step-by-step analysis, the judge should identify the first major error that caused the prediction to fail and create both a *summary* of a few sentences and an informative *label* that highlights the failed skill. We focus on the first major error because it often sets the trajectory for the rest of the reasoning; once an initial mistake is made, subsequent steps are likely to be flawed as well. This label is a phrasal description of the identified error. Finally, the judge outputs a JSON object that includes the necessary detailed criteria, along with the error summary and label. Note that, while only the error label is used in the next stage, the criteria and summary are helpful for interpreting each specific wrong prediction.

**Stage 2: Error Categorization** This stage consolidates instance-level results from the previous analysis into a list of common error types by iteratively grouping unique error labels into broader categories. Each category is assigned a description to reduce ambiguity.

To construct the categories, we adopted the data mining approach proposed by Wan et al. (2024), which iteratively employs an LLM to generate categories from input data, in our case, from the unique error labels and their prevalence. We summarize its 3 stages below (c.f., Wan et al., 2024), and provide in Appendix their prompts (§A.1.2, §A.1.3, §A.1.4) and configuration (Table A.1).

1. *Category Generation* – The initial stage, where the LLM receives the first batch (a list of error labels with their frequencies) and generates categories and category descriptions based on it. Note that since error labels from stage 1 were created in free-form, label repetitions were not guaranteed, though we observed frequent overlaps.

2. *Iterative Refinement* – Multiple iterations (depending on data size), where the LLM receives the previously generated categories along with a new sampled batch and incrementally updates and improves the categories.

3. *Final Review* – A concluding iteration where the LLM reviews the final taxonomy to ensure coherence and compliance with the instructions (e.g., no ambiguity).

The output of this stage is the final list of categories with their descriptions, produced after the review.

**Stage 3: Error Taxonomy Assignment** This stage integrates the outcomes of the previous two steps into a unified outcome. Specifically, we populate the taxonomy by assigning each instance-level full analysis (including the criteria analysis, error summary, and error label) to the most appropriate category, based on the classification of the error label.

| Error Type | Description |
| --- | --- |
| Calculation Error | Mistakes in arithmetic, algebraic manipulation, or numeric computation. |
| Reasoning Error | Flawed deduction, proof steps, or argumentative flow. |
| Incomplete Content | Omitting or partially providing essential elements demanded by the prompt (e.g., tables, explanations). |
| Constraint Violation | Breaking explicit instruction constraints such as word limits. |
| Language Issue | Misspellings, grammar mistakes, or language-specific constraint breaches. |
| Data Extraction | Failing to retrieve or include required numerical data or statistics. |
| Naming Error | Referring to the wrong entity, concept, or component. |
| Incorrect Method/Application | Using the wrong formula, theorem, or procedure for the problem. |
| Factual Error | Providing statements that are factually incorrect or outdated. |
| Formatting Error | Errors in markup, syntax, or incorrect format. |
| Question Misinterpretation | Misunderstanding the prompt's intent, leading to an incorrect approach. |
| Code Error | Mistakes in code snippets, function signatures, or programming logic. |
| Unwarranted Assumption | Adopting an unsupported premise that drives the solution. |
| Irrelevant/Off-Topic | Response does not address the question or task. |
| Verbosity | Excessive repeated content or length without adding value. |
| Policy Violation | Providing disallowed, unsafe, or prohibited content contrary to policy. |
| Refusal | Model refuses or gives a non-compliant answer. |
| Hallucination | Inventing references, data, or details that do not exist. |

Table 1: `ErrorAtlas`: High-level error categories and category descriptions.

This integration is done using a simple batched LLM call (the prompt is provided in App. §A.1.5). We provide the model with the error categories and a batch of error labels (see classify bath size parameter in Appendix, Table A.1), and ask the model to assign each error to the most appropriate category.

The outcome of this stage is a layered analysis. The top layer consists of the final categories in the taxonomy. Each category contains error labels, and each label groups instances with their error summaries, identified during the per-instance stage. For example, the category "Unwarranted Assumption" includes the error label "Incorrect Uniqueness Assumption" which in turn contains the error summary: *"The model incorrectly assumes that only a circle can satisfy the translation condition, failing to consider or construct a valid non-circular convex counter example."* Additional examples can be found in Appendix A.2.

## 3 CONSTRUCTING ERRORATLAS

Where `ErrorMap` supplies a flexible way to acquire a dedicated taxonomy for a nuanced issue, such as specific models or task data, a static taxonomy is often preferred in cases where the replicability and broad comparisons are required. `ErrorAtlas` is built to accommodate such error analysis use-cases. We describe `ErrorAtlas`, a taxonomy that categorizes failure modes commonly shown by current popular models, built using `gpt-oss-120b`. We detail the process of constructing `ErrorAtlas`, including practical decisions made, such as the identity of the datasets. We refer to experimental details that are general to all our experiments, from building the taxonomy to validating it in Appendix B.

**Coverage** To create `ErrorAtlas`, we select a diverse group of 21 datasets spanning a wide range of tasks, domains, and capabilities, and extract an `ErrorMap` taxonomy across all available model predictions. In total, we include predictions from 73 models. We cover the scope of LLM evaluations

| Use Case | Persona | Goal | Example |
|----------|---------|------|---------|
| Model Debugging | Model Developer | Identify regressions and behavioral changes | Compare model versions (e.g., v1 vs. v2) to detect reductions in specific error types, such as reasoning failures, especially when targeting improvements in those areas |
| Benchmark Analysis | Benchmark Creator | Reveal model capabilities, provide error distributions and debug dataset validity | Run `ErrorMap` on benchmark results to characterize model error distribution and error types across tasks |
| Model Selection | Product Team | Choose the most suitable model for deployment | Select the best model on domain-specific tasks based on stakeholder preferences (e.g., prioritizing fewer hallucinations in medical applications). |
| Domain-Specific Evaluation | Domain Expert / Analyst | Identify failure modes in specialized contexts | Use `ErrorMap` to analyze model responses in high-stakes domains (e.g., legal, medical) and surface common failure patterns |

Table 2: Summary of key use cases using `ErrorMap`.

with the following benchmarks: from HELM leaderboards (Liang et al., 2023), Capabilities for general capabilities (Xu et al., 2024), MedHELM for medical domain (Bedi et al., 2025) and ToRR for tables (Ashury-Tahan et al., 2025), and for code HumanEval (Chen et al., 2021), HumanEval Plus (Liu et al., 2023b), MBPP (Austin et al., 2021) and MBPP Plus (Liu et al., 2023b). For both ToRR and MedHELM, we selected partial subsets of the datasets they contain.[2]

**Scaling `ErrorMap` through Sampling** The flexible nature of `ErrorMap` allows it to be applied to any number of models, datasets, and incorrect predictions. However, constructing `ErrorAtlas`, a unified taxonomy across all mentioned benchmarks, is far more demanding than applying `ErrorMap` in narrow settings (e.g., a single benchmark or a small set of models). The number of errors to analyze scales with the dataset sizes and number of models, resulting in significant computational demands, a common obstacle for large scale evaluations (Perlitz et al., 2024). To manage the data volume, we employed relative sampling: for each model-dataset pair, we sampled approximately 10% of the instances where the model was evaluated as having failed, i.e., a proportionate subset based on the model's error rate. This resulted in a sample of over 7,000 failures that was then used to run `ErrorMap`. Interestingly, despite the sample size and its variability, no duplicate categories were observed in the resulting taxonomy. This may suggest that the iterative refinement effectively consolidates similar error types and is not sensitive to sample size.

**Manual Taxonomy Refinement** Applying `ErrorMap` to the sampled data resulted in a taxonomy with interpretable categories. However, manual inspection suggested many of the generated labels were overly verbose, for example, labels like "Refusal / Non-compliant Response". We therefore manually refined the high-level categories while preserving their original semantics. To ensure generality, we filtered out categories present in fewer than 20% of datasets or deemed rare/uninformative. Further details and the original taxonomy are provided in App. Table C.2.

**Usage** Now that we have extracted `ErrorAtlas`, its primary value is in clearly surfacing common LLM error types (see §4). This can support future model development and real-world improvements, particularly as we uncover previously unreported error categories. Moreover, `ErrorAtlas` categories can be practically applied at low cost to reflect general model failure modes. This can be done by running only Stage 1 and Stage 3, while skipping Stage 2 (Error Categorization).[3]

---

[2]We focused on tasks where model outputs include interpretable content, as `ErrorMap` goal is to analyze predictions that reveal failure modes. Many benchmark tasks, like classification or entity extraction do not provide explanations (or CoT) and lack the necessary context for such analysis. The full details of selected datasets are provided in App. Table C.1

[3]Running stage 2 on specific data may be less representative from a model's general failure mode perspective, as it depends on data collection that may be biased.

## 4 ERRORATLAS APPLICABILITY

**ErrorAtlas Reveals the Error Topography of Models** Running `ErrorMap` on 21 datasets results in the construction of `ErrorAtlas` (see §3). The main outcome is a set of 18 high-level taxonomy categories describing common model errors, presented in Table 1. Examples of error categories with their children labels are shown in Figure 1 and the original resulted taxonomy with statistics is available in App. Table C.3.

The resulting error categories span a wide spectrum, reflecting diverse dimensions of model performance. These include reasoning-related errors, such as Reasoning Errors, Unwarranted Assumptions, Naming Errors, and Question Misinterpretation; instruction-following issues, including Constraint Violations, Policy Violation and Incomplete Content; procedural errors, such as Calculation Errors, Incorrect Method/Application, and Data Extraction Failures; and technical and linguistic issues, including Language Problems, Verbosity, and Formatting Errors. Additionally, there are categories that fall outside these dimensions, such as Refusal.

While the areas of failure described above (e.g., instruction following and reasoning) are generally well-known and researched within the community, `ErrorAtlas` enables the identification of more precise weaknesses within these broader categories. Moreover, there resulted taxonomy underscores a key limitation of benchmark scores: although they provide useful indicators of model performance on specific tasks or domains, they often lack the granularity required to uncover detailed failure patterns. Understanding these patterns is essential for diagnosing concrete limitations in model behavior and guiding targeted improvements.

**Surfacing Frequent but Overlooked Model Failures** While some error types are more actively studied, such as reasoning errors (Zheng et al., 2025; Xu et al., 2025; Liu et al., 2023a) and hallucinations (Cattan et al., 2025; Zhao et al., 2024) and others can be mitigated through techniques like tool use (e.g., resolving calculation mistakes), `ErrorAtlas` reveals additional error patterns that have received limited attention in the community, despite their prevalence not justifying such disproportionate neglect.

One such pattern, is labeled in `ErrorAtlas` as *Incomplete Content*. Surprisingly, despite being the most prevalent error in our analysis, this error type is under-discussed[4]. Upon manual inspection of the results, we found that this pattern usually involves missing details with respect to the context, such as not fully answering the question, omitting specific nuances requested, or ignoring certain instructions and constraints. Examples provided in Appendix C.1.1 show such cases where the model produces a partially correct solution. This tendency to overlook contextual cues can significantly impact the reliability of AI systems. For instance, while a set of symptoms may typically suggest a particular diagnosis, subtle nuances in a specific case could point to a completely different one.

Another error shown in Tables 1,C.3, with notable prevalence across datasets, is *Question Misinterpretation*. Examining specific instances in this category reveals various cases where models fail to adequately consider context or respond with

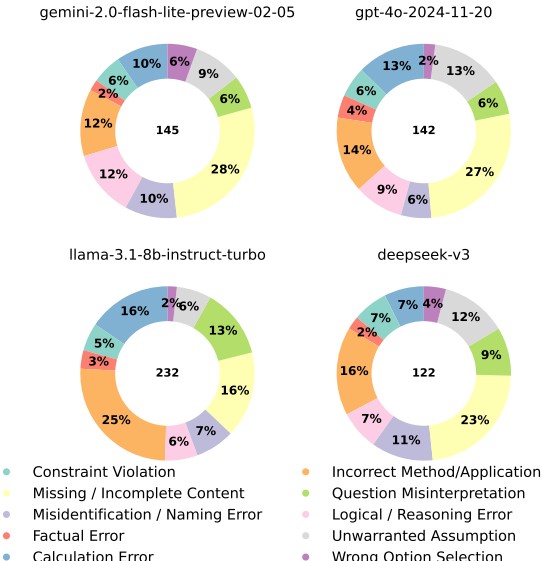

Figure 2: Error distributions for models across the 10 most prevalent `ErrorAtlas` error categories on the Capabilities benchmark. The total number of errors for each model appears at the center of its corresponding donut chart. Each error count represents roughly 10% of the model's total errors on Capabilities.

---

[4]Similar issues were hardly mentioned in related work search and existing taxonomies, with the exception or retrieval literature.

the required expertise. These include instances of misalignment between surface cues and deeper context, as well as failures to interpret the information provided in the context (see examples in Appendix C.1.2). This underscores the need for improved contextual understanding in model development, particularly for tasks requiring nuanced interpretation.

**Error Patterns Vary Between Models.**

Employing `ErrorAtlas`, we observe distinct model-specific patterns that reveal nuanced variations in error behavior. To quantify these differences, we analyzed the error distributions of models. To ensure a fair comparison, we selected models that appear in the same benchmark, HELM Capabilities, and measured their error distributions within it. Figure 2 illustrates the top 10 error categories across several models. Notably, all models tend to make errors related to incomplete content, with `Gemini 2.0 Flash Lite` exhibiting the highest frequency in this category, while showing the fewest errors in incorrect method/application, a pattern reversed in `LLaMA 3.1 8B`. `DeepSeek V3` shows a higher tendency toward naming errors compared to others, whereas `GPT-4o` is more prone to unwarranted assumptions.

Moreover, it appears that models with a higher number of errors are more prone to making mistakes related to incorrect method or application, whereas models with fewer errors tend to struggle more with incomplete content. Interestingly, there doesn't seem to be a strong signal related to reasoning errors. Supporting figures illustrating these patterns can be found in Appendix C.

**Model Error Does not Always Reflect Failure in the Benchmark Targeted Skill.** We motivated our reliance on model outputs by the discrepancy between what a question aims to test and what eventually trips the model. We indeed find such cases in our analysis. For example, consider capability-focused datasets like MMLU-Pro, Omni-MATH, and GPQA, which are considered challenging due to their reasoning demands. While Omni-MATH emphasizes math reasoning, GPQA focuses on general reasoning, and MMLU-Pro primarily tests knowledge along with a reasoning depth. However, approximately 47% of model errors in these benchmarks have a weak reasoning orientation, and seem more technical challenges, e.g., calculation error, incorrect application, or missing content (see App. Table C.4).

Overall, we have shown the usefulness of `ErrorAtlas` for comparing model providers, models, and gaining insight in a new domain. In §5, we discuss the cases where a dedicated taxonomy is helpful and showcase it.

## 5 ERRORMAP APPLICABILITY

In Section 4 we saw `ErrorAtlas` enables evaluations that help track model weaknesses and monitor progress over time when rerun. However, many diagnostic users are interested in their specific failure modes rather than general ones. Table 2 summarizes five key use cases, each illustrating how `ErrorMap` can aid decision-making, debugging, and evaluation in various contexts. In this section we demonstrate the first and second use-cases, aimed at supporting model developers and benchmark persona. The experimental setup is provided in App. B.

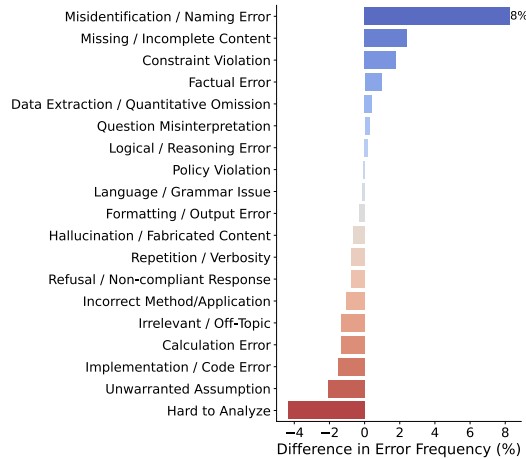

Figure 3: Differences in error frequency between `Gemini 1.5` flash and Pro on the capabilities benchmark in HELM. X-axis represents the change in error frequency, highlighting areas of improvement or regression.

### 5.1 MODEL DEVELOPERS

For model developers, `ErrorMap` provides a structured way to assess behavioral changes between iterations (similar to a behavioral model-diff; Mishra-Sharma et al., 2025; Lindsey et al., 2024; Aranguri & McGrath, 2025). For example, it can help answer questions like: What common errors in my setting did my improved version ad-

dress? or Did integration with external tools reduce hallucinations? By surfacing such differences, `ErrorMap` supports more informed and targeted improvements.

To test this, we compared `gemini-1.5-flash` and `gemini-1.5-pro` using capabilities benchmark data from HELM. While the pro version outperforms the flash version by a mean score of $4.8\%$ on the benchmark, one may wonder what are the differences between them. Our analysis (shown in Figure 3) presents the differences in the percentage of

| ErrorMap Categories | MMLU-Pro Paper Categories |
|---|---|
| Logical Reasoning Error (44%) | Reasoning Errors (39%) |
| Mathematical Mistake (24%) | Calculation Errors (12%) |
| Incomplete Answer (13%) | Lack of Specific Knowledge (35%) |
| Factual Error (12%) | |
| Prompt Misinterpretation (5%) | Question Understanding Errors (4%) |
| | Other (10%) |

Table 3: Comparison of MMLU-Pro error categories and GPT-4o distribution between manual annotations from the original paper and our method.

their errors. It is evident that the pro model performs significantly less naming errors and is more capable of referring to the correct entity or concept in the question. A model developer can use such analysis to determine whether the changes made in the pro version were focused on that aspect, or are those unexpected changes that call for more developmental efforts.

We also note that this analysis can be valuable for other model stakeholders, helping them *make more informed decisions*. For example, if one has the budget to utilize Gemini models but seeks to optimize costs, this evaluation can serve as a valuable guide. In scenarios where the task involves precise identification of entities, such as names in data extraction, the pro version may be preferable due to its enhanced capabilities. Conversely, for tasks primarily focused on summarization, where such precision is less critical, paying more for the pro version might not be cost-effective.

## 5.2 BENCHMARK CREATORS

Diagnosis is also important for benchmark curators to validate what key challenges it poses for models and highlight unexpected errors. To demonstrate usability, we use MMLU-Pro as a case study and apply `ErrorMap` to generate its taxonomy, demonstrating several key capabilities of our approach:

**`ErrorMap` closely approximates manual analysis in MMLU-Pro.** Running `ErrorMap` on MMLU-Pro dataset provided us with 5 error categories. We compared the manual analysis the paper reported for `gpt4o` with ours and got a similar error distribution (see Table 3), with the exception of two categories in `ErrorMap` that map to one manual one and no "other" category.

**Comparing Dataset Parts for Richer Interpretive Insights** While `ErrorMap` separates analysis bottom up by the errors, integrating high-level data dimensions can yield more nuanced results. To demonstrate this, we analyzed model errors in relation to the domain categories of the dataset, as shown in App. Figure D.1. Some patterns appear intuitive, for example, mathematics and physics exhibit similar error distributions. However, other findings are less expected, such as the disproportionately high number of factual errors in the health domain, even exceeding those in history.

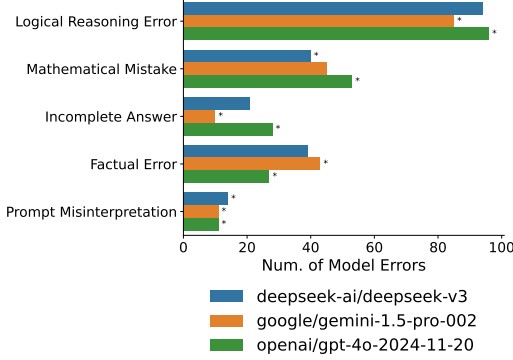

Figure 4: Differences in error category distributions among three leading models on the MMLU-Pro dataset. Asterisks (*) indicate the bars that were compared in the statistical significance test.

| Error Category | P-val ($\downarrow$) |
|---|---|
| Factual Error | .000218 |
| Incomplete Answer | .000000 |
| Logical Reasoning Error | .000333 |
| Mathematical Mistake | .002563 |
| Prompt Misinterpretation | .074530 |

Table 4: Significance testing between the best- and worst-performing models for each error category. The results show that differences between models are usually statistically significant.

We further note that benchmark users can benefit from the model comparisons presented in benchmarks to distinguish between models in a more granular fashion. To demonstrate this, we compare three models on their MMLU-Pro error distributions in Figure 4. For instance, we find that GPT-4o exhibits a higher proportion of reasoning errors compared to Gemini 1.5 Pro, which, in contrast, makes significantly more factual errors.

In conclusion, ErrorMap enhances understanding of benchmark datasets beyond overall metrics. By revealing task-specific insights and taxonomy, it helps users interpret model behavior and complements leaderboard reporting with deeper context.

# 6 VALIDATING ERRORMAP

In §5, we previously demonstrated the utility of ErrorMap; we now evaluate whether its components function as expected, with a particular emphasis on the resulting taxonomy, which we consider a key contribution of this work. The following evaluations include quantitative ones conducted using Qwen2.5-72B-Instruct as a meta-judge, along with qualitative ones performed manually, detailed in Appendix E.

## 6.1 PER-INSTANCE ERROR ANALYSIS

Two key aspects of the per-instance analysis stage are accuracy and robustness. While accuracy measures whether the judge assigned a correct label to the error, robustness can be evaluated in multiple ways (Habba et al., 2025). In this work, we adopt a commonly used approach to measure robustness by examining the model's sensitivity to prompt variations (Pezeshkpour & Hruschka, 2023; Errica et al., 2024; Zhuo et al., 2024).

To evaluate *accuracy*, we provided the meta-judge with all the information given to the original judge and its proposed analysis. Notably, the meta-judge's task is significantly simpler than that of the analysis component. While the judge must generate a coherent explanation for the error, often requiring reasoning across multiple steps, the meta-judge only verifies whether the given analysis correctly explains the error. In other words, the judge performs a binary classification (correct/incorrect) based on a predefined context, without needing to produce or synthesize new information. The meta-judge accepted the vast majority of instances in each case with an average score of **91.1%** (see details in Appendix E).

For assessing *robustness*, we followed the approach of Kamoi et al. (2024), executing the per-instance stage with 3 prompt variations. We then compared the consistency of the 2 resulting error analyses with the original prompt setup. This comparison is challenging to automate, as the error labels are free-form and may differ in non-relevant ways (e.g., style or level of generality). To approximate a quantitative measure, we computed pairwise cosine similarity between error labels from the original and varied prompts using Sentence-BERT embeddings (Reimers & Gurevych, 2019)(details in App.B). The average similarity score was moderate **53%**. we analyzed manually 100 examples, and found them to be divided into the following categories: 45% included the same underlying concept, but one phrase was partial to another in the text. Another 30% suffered from varying specificity in the labels and not disagreements. For instance, the labels "missing temporal specification for midnight setting" and "missing explicit time reference" received a similarity score of 0.21. And 25% included a different error, and this may stem from the soft nature of errors, that may be called in multiple ways.

## 6.2 ERROR TAXONOMY

Building on prior work (Wan et al., 2024; Shah et al., 2023), we evaluate the taxonomy using three main criteria: *coverage*, how comprehensively the taxonomy captures error types; *accuracy*, how reliably it categorizes them, and *usefulness*, how well it aligns with the intended application. Usefulness is central to the taxonomy's practical value, reflecting its support for downstream tasks; we therefore dedicate Section §5 to this aspect. Additionally, we introduce robustness as a fourth criterion, motivated by concerns in the literature about the reliability of LLM-based evaluations (Mizrahi et al., 2024; Siska et al., 2024; Lior et al., 2025). Robustness measures the stability of the taxonomy under variations in prompt phrasing or evaluator perspective.

Taxonomy *coverage* is evaluated using the approach proposed by Wan et al. (2024). For each sampled error instance, we attempt to automatically map it into the taxonomy. If no suitable category is found, we assign it to an "other" category. We further added "hard to analyze" category for cases where there is not enough information to analyze the example. As shown in Table C.3, only 1 example was classified into the "other" category, and 48 were classified into the "hard to analyze". We further include rare or uninformative categories, those not part of ErrorAtlas, as part of this group, totaling 295 errors. With 7,049 analyzed wrong predictions, we obtain a coverage score of **95.2%**.

We measure taxonomy *accuracy* using the approach of Wan et al. (2024). In each evaluation round, we sample an error instance from the taxonomy and present it to a judge model. The judge is given the instance's assigned label from the taxonomy, along with an alternative negative label from it at random. Based on this information, the judge is asked to determine which label better fits the error. The results of this evaluation are shown in App. Table E.1 and indicate a high level of agreement with the assigned labels, averaging an accuracy score of **92%**.

We assess the *robustness* of our taxonomy by comparing the categories produced by ErrorAtlas with those obtained from 2 additional runs, a different sample of the data (using a different seed and sample ratio 5%, 15%), and a prompt that paraphrases some of the original instructions. This comparison resulted in a highly similar category list, as shown in App. Table E.2. Specifically, 21 categories were found to be semantically equivalent, and an additional 4 categories were more nuanced and not present in the perturbed version. Overall, the results indicate a strong overlap and consistency across variations.

# 7 RELATED WORK

While various works studied common errors in a specific setups, such as errors within a particular subdomain (Dou et al., 2024; Wang et al., 2024b; Ramprasad et al., 2024; Deshpande et al., 2025) or errors specific to a single model (Yehudai et al., 2025), or have defined challenges through the lens of question difficulty (Bradley, 2024; Baldock et al., 2021; Hacohen et al., 2020; Choshen et al., 2022; Habba et al., 2025), we are aware of no work that presents a general LLM error taxonomy or explores the use of global model error signals. Taxonomies by latent skills were used in contexts such as scaling laws (Polo et al., 2024) and self-specialization (Kang et al., 2023) to find, possibly non-interpretable, dimensions that describe shared model performance. Maimon et al. (2025) utilize latent skills for diagnosis, and create a static dedicated leaderboard to act as an IQ test for LLMs. Some manual efforts split input data to highlight instances posing a shared challenge (e.g., Magnusson et al., 2023), and others automate such practices (Choshen & Abend, 2019; Tjuatja & Neubig, 2025). Other works extract what each input tests to analyze model outputs (Zeng et al., 2025) or predict their skills (Zhou et al., 2025; Ruan et al., 2024; Polo et al., 2024). We see great value in such works. While we explain the benefits of analyzing actual errors and relying on model outputs rather than the challenges in the inputs, we believe these works to be useful for different needs.

We also note that a recurring part driving the decisions made in ErrorMap is efficiency. Currently, as it runs only on failed examples and batches it takes a similar amount of inference as the evaluation, in ErrorAtlas we also sample. A large body of works suggested ways of efficient sampling for evaluation, they also consistently find that for most cases a random sample is a strong baseline and most alternatives introduce a bias (Choshen et al., 2024; Zhuang et al., 2025; Wang et al., 2025; Polo et al., a;b; Maia Polo et al., 2024; Perlitz et al., 2024). Since none of those methods aim for a distribution of errors and analysis, we sampled randomly.

# 8 CONCLUSIONS

In this work, we presented ErrorMap, a diagnosis method, to efficiently produce a summary of the errors models perform on a given benchmark, enabling more interpretable comparisons between models. We also introduced ErrorAtlas, a general taxonomy emphasizing current LLM errors, contributing to a deeper understanding of their behaviors. While obtaining a complete picture is an inherently challenging goal, ErrorAtlas offers a first glimpse into current model errors, based on a diverse set of benchmarks spanning multiple domains and skills. Our approach lays the foundation for more detailed model evaluation. We refer to the limitations of our work to App. F.

## 9 ETHICS STATEMENT

We see diagnosis work as a net good. There are little obvious harms from better understanding where systems that we currently develop fail, except in aiding development of systems that shouldn't be developed in the same way it aids regular development. Importantly, our method supports responsible development, deployment, and auditing of models (see Table 2, particularly the domain expert persona).

## 10 REPRODUCIBILITY STATEMENT

Many efforts were done to make this work reproducible, at the end of the day, evaluation only matters if someone is using it. We will share the code upon acceptance. In addition we report all prompts used and hyperparameters of the methods in the appendices and the experimental setup and list all models and datasets used as well. We further mention that the method is designed to be robust, and rerunning the pipeline should yield similar results. However, as the approach relies on LLMs, inherent stochasticity and non-determinism may lead to slight variations in outputs across runs.

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

# A ERRORMAP: ADDITIONAL DETAILS

| Field Name | Default Value | Description |
|---|---|---|
| batch size | 500 | Size of minibatches for data processing. |
| classify batch size | 50 | Size of minibatches for item classification. |
| cluster name length | 5 | Maximum length for cluster names. |
| cluster description length | 30 | Maximum length for cluster descriptions. |
| max num clusters | 25 | Maximum number of clusters allowed. |

Table A.1: Taxonomy Configuration Parameters.

## A.1 PROMPTS

### A.1.1 PER-INSTANCE ERROR ANALYSIS

```
{% set reasoning_effort = "high" %}

You are an expert analyst. Your job is to evaluate evidence step
    by step, consider alternatives, and reach a justified
    conclusion.

You are given the following:
- A context
- A model response that was labeled incorrect
{% if correct_answer %}
- A reference
{% endif %}
{% if correct_outputs %}
- A list of solutions that were labeled as correct
{% endif %}

Your task:

1. Structured Correct Solution: Analyze the correct responses and
    extract from them the main required criteria or reasoning
    steps for the context.

2. Step-by-step Evaluation: Evaluate the incorrect response
    against each of the required criteria. For each criterion,
    provide the following fields:
present_in_wrong: Whether it is present in the incorrect response
quality: The quality of its execution (correct, partially correct,
    incorrect, or null if missing)
evidence: Supporting evidence from the incorrect response (quote)
comment: Any relevant comments

3. Error Diagnosis: Identify the first major error in the
    incorrect response that led to the incorrect answer, and
    provide the following fields in final_answer:
error_summary: If such an error exists, summarize the model's
    reasoning weakness in error_summary. This should focus on
    model thinking (e.g., 'the model failed to recognize fact X')
    rather than technical execution (e.g., 'the model selected the
    wrong answer').
title: Provide a short, free-form title that describes the
    specific type of error.
* If you didn't find any error in the incorrect response leave all
    the fields of final_answer with an empty string.
```

```
* If the whole solution is incorrect, write 'whole solution
    incorrect' in final_answer fields.
* Avoid ambiguous titles or ones that cannot be mapped to a
    specific skill. For example, instead of using "Wrong multiple
    choice selection", identify the underlying reasoning error
    such as "Misinterpretation of concept".

Use as many steps and thinking process as you need. Finally,
    output the final result in the following format:

{
  "required_criteria": [
    {
      "criterion": "Describe the relationship between A and B",
      "present_in_wrong": true,
      "quality": "incorrect",
      "evidence": "Because A increased when B increased, A must be
            caused by B.",
      "comment": "Confuses correlation with causation"
    },
    {
      "criterion": "Explain the mechanism of action",
      "present_in_wrong": true,
      "quality": "correct",
      "evidence": "the biochemical pathway...",
      "comment": "Accurate and complete"
    }
  ],
  "final_answer": {
    "error_summary": "The incorrect response assumes causation
        from correlation, leading to a flawed conclusion about the
         relationship between A and B.",
    "error_title": "Causal Misinterpretation"
  }
}

Use the following inputs:

Context:
{{ input_text }}

{% if candidate_answers %}
Candidate Answers:
{{ candidate_answers }}
{% endif %}

{% if correct_answer %}
References:
{{ correct_answer }}
{% endif %}

{% if correct_outputs %}
Correct Responses:
{{ correct_outputs }}
{% endif %}

incorrect prediction:
{{ output_text }}
```

```
Keeping the evaluation criteria in mind, do not provide a general
    assessment. Be specific, structured, and evidence-based.

Assessment:
```

### A.1.2  TAXONOMY GENERATION PROMPT

```
{% set reasoning_effort = "high" %}

You are an expert analyst. Your job is to evaluate evidence step
    by step, consider alternatives, and reach a justified
    conclusion.

# Instruction
## Context
- **Goal**: Your goal is to cluster the input data into meaningful
     categories for the given use case.
- **Data**: The input data will be a list of {{ data_type }}
    tuples, including the following elements:
    - **text**: {{ data_type }} as the first tuple element.
    - **num of occurrences**: number as the second tuple element.
- **Use case**: Generate a taxonomy that categorizes model errors
    based on the specific skills the model failed to demonstrate
    in each example.

## Requirements

### Format
- Output clusters in **XML format** with each cluster as a '<
    cluster >' element, containing the following sub-elements:
  - **id**: category number starting from 1 in an incremental
      manner.
  - **name**: category name should be **within {{
      cluster_name_length }} words**. It can be either verb phrase
      or noun phrase, whichever is more appropriate.
  - **description**: category description should be **within {{
      cluster_description_length }} words**.

Here is an example of your output:
```xml
<clusters>
  <cluster>
    <id>category id</id>
    <name>category name</name>
    <description>category description</description>
  </cluster>
</clusters>
```

- Total number of categories should be **no more than {{
    max_num_clusters }}**.
- Output should be in **English** only.

### Quality

- **No overlap or contradiction** among the categories.
```

- **Name** is a concise and clear label for the category,
  identifies **one specific skill or ability only**. Use only
  phrases that are specific to each category and avoid those
  that are common to all categories.
- **Name** reflects core capabilities, not domain−specific
  contexts, or technical choices.
  Example: not "Incorrect Anatomical Knowledge" but "Factual Error
  " (The issue is about factual accuracy, not biology
  specifically).
  If the issue does not clearly map to a specific skill, classify
  it as "Hard to Analyze" − this applies when the error is
  ambiguous, subjective, or lacks sufficient context to
  determine its nature confidently.
- **Description** differentiates one category from another.
- **Name** and **description** can **accurately** and **
  consistently** classify new data points **without ambiguity**.
- **Name** and **description** are *consistent with each other*.
- Output clusters match the data as closely as possible, without
  missing important categories or adding unnecessary ones.
- Output clusters should strive to be orthogonal, providing solid
  coverage of the target domain.
- Output clusters serve the given use case well.
- Output clusters should be specific and meaningful. Do not invent
  categories that are not in the data.

# Data
<{{ data_type }}>
{{ data }}
</{{ data_type }}>

# Questions
## Q1. Please generate a cluster table from the input data that
  meets the requirements.

Tips

- **User Feedback is MANDATORY**: You MUST address any previous
  user feedback in your clustering
- If user feedback was provided, explicitly explain how you've
  incorporated their specific concerns and suggestions
- The cluster table should be a **flat list** of **mutually
  exclusive** categories. Sort them based on their semantic
  relatedness.
- Though you should aim for {{ max_num_clusters }} categories, you
  can have *fewer than {{ max_num_clusters }} categories* in
  the cluster table; but **do not exceed the limit.**
- Be **specific** about each category. **Do not include vague
  categories** such as "Other", "General", "Unclear", "
  Miscellaneous" or "Undefined" in the cluster table.
- You can ignore low quality or ambiguous data points.

## Q2. Why did you cluster the data the way you did? Explain your
  reasoning **within {{ explanation_length }} words**. Include
  how you addressed any user feedback.

## Provide your answers between the tags: <cluster_table>your
  generated cluster table with no more than {{ max_num_clusters
  }} categories</cluster_table>, <explanation>explanation of

```
your reasoning process within {{ explanation_length }} words</
    explanation >.

# Output
```

A.1.3   TAXONOMY UPDATE PROMPT

```
{% set reasoning_effort = "high" %}

You are an expert analyst. Your job is to evaluate evidence step
    by step, consider alternatives, and reach a justified
    conclusion.

# Instruction
## Context
- **Goal**: You goal is to review the given reference table based
    on the input data for the specified use case, then update the
    reference table if needed.
    - You will be given a reference cluster table, which is built
        on existing data. The reference table will be used to
        classify new data points.
    - You will compare the input data with the reference table,
        output a rating score of the quality of the reference
        table, suggest potential edits, and update the reference
        table if needed.
- **Reference cluster table**: The input cluster table is in XML
    format with each cluster as a '<cluster>' element, containing
    the following sub-elements:
    - **id**: category index.
    - **name**: category name.
    - **description**: category description used to classify data
        points.
- **Data**: The input data will be a list of {{ data_type }}
    tuples, including the following elements:
    - **text**: {{ data_type }} as the first tuple element.
    - **num of occurrences**: number as the second tuple element.
- **Use case**: Update the taxonomy that categorizes model errors
    based on the specific skills the model failed to demonstrate
    in each example.

## Requirements

### Format
- Output clusters in **XML format** with each cluster as a '<
    cluster>' element, containing the following sub-elements:
    - **id**: category number starting from 1 in an incremental
        manner.
    - **name**: category name should be **within {config.
        cluster_name_length} words**. It can be either verb phrase
        or noun phrase, whichever is more appropriate.
    - **description**: category description should be **within {
        config.cluster_description_length} words**.

Here is an example of your output:
```xml
<clusters>
  <cluster>
    <id>category id</id>
```

```
      <name>category name</name>
      <description>category description</description>
    </cluster>
</clusters>
```

- Total number of categories should be **no more than {config.
  max_num_clusters}**.
- Output should be in **English** only.

### Quality

- **No overlap or contradiction** among the categories.
- **Name** is a concise and clear label for the category,
  identifies **one specific skill or ability only**. Use only
  phrases that are specific to each category and avoid those
  that are common to all categories.
- **Name** reflects core capabilities, not domain-specific
  contexts, or technical choices.
  Example: not "Incorrect Anatomical Knowledge" but "Factual Error
  " (The issue is about factual accuracy, not biology
  specifically).
  If the issue does not clearly map to a specific skill, classify
  it as "Hard to Analyze" - this applies when the error is
  ambiguous, subjective, or lacks sufficient context to
  determine its nature confidently.- **Description**
  differentiates one category from another.
- **Name** and **description** can **accurately** and **
  consistently** classify new data points **without ambiguity**.
- **Name** and **description** are *consistent with each other*.
- Output clusters match the data as closely as possible, without
  missing important categories or adding unnecessary ones.
- Output clusters should strive to be orthogonal, providing solid
  coverage of the target domain.
- Output clusters serve the given use case well.
- Output clusters should be specific and meaningful. Do not invent
  categories that are not in the data.

# Reference cluster table
<reference_table>
{{ cluster_table_xml }}
</reference_table>

# Data
<{{ data_type }}>
{{ data }}
</{{ data_type }}>
# Reference cluster table

# Questions
## Q1: Review the given reference table and the input data and
  provide a rating score of the reference table. The rating
  score should be an integer between 0 and 100, higher rating
  score means better quality. You should consider the following
  factors when rating the reference cluster table:
- **Intrinsic quality**:
  - 1) if the cluster table meets the *Requirements* section,
    with clear and consistent category names and descriptions,
    and no overlap or contradiction among the categories;

```
        - 2) if the categories in the cluster table are relevant to
            the the given use case;
        - 3) if the cluster table includes any vague categories such
            as "Other", "General", "Unclear", "Miscellaneous" or "
            Undefined".
- **Extrinsic quality**:
    - 1) if the cluster table can accurately and consistently
        classify the input data without ambiguity;
    - 2) if there are missing categories in the cluster table but
        appear in the input data;
    - 3) if there are unnecessary categories in the cluster table
        that do not appear in the input data.

## Q2: Explain your rating score in Q1 **within {{
    explanation_length }} words**.

## Q3: Based on your review, decide if you need to edit the
    reference table to improve its quality. If yes, suggest
    potential edits **within {{ suggestion_length }} words**. If
    no, please output the original reference table.

Tips:
- You can edit the category name, description, or remove a
    category. You can also merge or add new categories if needed.
    Your edits should meet the *Requirements* section.
- The cluster table should be a **flat list** of **mutually
    exclusive** categories. Sort them based on their semantic
    relatedness.
- You can have *fewer than {{ max_num_clusters }} categories* in
    the cluster table, but **do not exceed the limit.**
- Be **specific** about each category. **Do not include vague
    categories** such as "Other", "General", "Unclear", "
    Miscellaneous" or "Undefined" in the cluster table.
- You can ignore low quality or ambiguous data points.

## Q4: If you decide to edit the reference table, please provide
    your updated reference table. If you decide not to edit the
    reference table, please output the original reference table.

## Provide your answers between the following tags:
<rating_score>integer between 0 and 100</rating_score>
<explanation>explanation of your rating score within {{
    explanation_length }} words</explanation>
<suggestions>suggested edits within {{ suggestion_length }} words,
    or "N/A" if no edits needed</suggestions>
<updated_table>
your updated cluster table in XML format if you decided to edit
    the reference table, or the original reference table if no
    edits made
</updated_table>
# Output
```

A.1.4  TAXONOMY REVIEW PROMPT

```
{% set reasoning_effort = "high" %}
```

```
You are an expert analyst. Your job is to evaluate evidence step
    by step, consider alternatives, and reach a justified
    conclusion.

# Instruction
## Context
- **Goal**: Your goal is to review the given reference table based
    on the requirements and the specified use case, then update
    the reference table if needed.
    - You will be given a reference cluster table, which is built
        on existing data. The reference table will be used to
        classify new data points.
    - You will compare the reference table with the requirements,
        output a rating score of the quality of the reference
        table, suggest potential edits, and update the reference
        table if needed.
- **Reference cluster table**: The input cluster table is in XML
    format with each cluster as a '<cluster>' element, containing
    the following sub-elements:
    - **id**: category index.
    - **name**: category name.
    - **description**: category description used to classify data
        points.
- **Use case**: Review the taxonomy that categorizes model errors
    based on the specific skills the model failed to demonstrate
    in each example.

## Requirements

### Format
- Output clusters in **XML format** with each cluster as a '<
    cluster>' element, containing the following sub-elements:
  - **id**: category number starting from 1 in an incremental
      manner.
  - **name**: category name should be **within {{
      cluster_name_length }} words**. It can be either verb phrase
       or noun phrase, whichever is more appropriate.
  - **description**: category description should be **within {{
      cluster_description_length }} words**.

Here is an example of your output:
```xml
<clusters>
  <cluster>
    <id>category id</id>
    <name>category name</name>
    <description>category description</description>
  </cluster>
</clusters>
```

- Total number of categories should be **no more than {{
    max_num_clusters }}**.
- Output should be in **English** only.

### Quality

- **No overlap or contradiction** among the categories.
```

```
– **Name** is a concise and clear label for the category,
  identifies **one specific skill or ability only**. Use only
  phrases that are specific to each category and avoid those
  that are common to all categories.
– **Name** reflects core capabilities, not domain-specific
  contexts, or technical choices.
  Example: not "Incorrect Anatomical Knowledge" but "Factual Error
      " (The issue is about factual accuracy, not biology
      specifically).
  If the issue does not clearly map to a specific skill, classify
      it as "Hard to Analyze" – this applies when the error is
      ambiguous, subjective, or lacks sufficient context to
      determine its nature confidently.– **Description**
      differentiates one category from another.
– **Name** and **description** can **accurately** and **
  consistently** classify new data points **without ambiguity**.
– **Name** and **description** are *consistent with each other*.
– Output clusters match the data as closely as possible, without
  missing important categories or adding unnecessary ones.
– Output clusters should strive to be orthogonal, providing solid
  coverage of the target domain.
– Output clusters serve the given use case well.
– Output clusters should be specific and meaningful. Do not invent
    categories that are not in the data.

# Reference cluster table
<reference_table>
{{ cluster_table_xml }}
</reference_table>

# Questions
## Q1: Review the given reference table and provide a rating score
    . The rating score should be an integer between 0 and 100,
    higher rating score means better quality. You should consider
    the following factors when rating the reference cluster table:
    – **Intrinsic quality**:
        – If the cluster table meets the required quality with
            clear and consistent category names and descriptions,
            and no overlap or contradiction among the categories.
        – If the categories in the cluster table are relevant to
            the specified use case.
        – If the cluster table does not include any vague
            categories such as "Other", "General", "Unclear", "
            Miscellaneous" or "Undefined".
    – **Extrinsic quality**:
        – If the cluster table can accurately and consistently
            classify the input data without ambiguity.
        – If there are missing categories in the cluster table
            that appear in the input data.
        – If there are unnecessary categories in the cluster table
            that do not appear in the input data.

## Q2: Explain your rating score in Q1 [The explanation should be
    concise, based on the intrinsic and extrinsic qualities
    evaluated in Q1].

## Q3: Based on your review, decide if you need to edit the
    reference table to improve its quality. If yes, suggest
    potential edits [Suggestions should be specific, actionable,
```

```
and within the constraints of the maximum number of categories
    and use case specificity].

## Q4: If you decide to edit the reference table, provide your
    updated reference table. If you decide not to edit the
    reference table, please output the original reference table.

## Provide your answers between the following tags:
<rating_score>integer between 0 and 100</rating_score>
<explanation>concise explanation of your rating score based on the
    intrinsic and extrinsic qualities</explanation>
<suggestions>specific and actionable suggestions for edits, or "N/
    A" if no edits needed</suggestions>
<updated_table>
your updated cluster table in XML format if you decided to edit
    the reference table, or the original reference table if no
    edits made
</updated_table>

# Output
```

### A.1.5 ERROR LABEL CLASSIFICATION PROMPT

```
{% set reasoning_effort = "high" %}

You are an expert analyst. Your job is to evaluate evidence step
    by step, consider alternatives, and reach a justified
    conclusion.

Your task is to use the provided taxonomy to categorize the
    overall topic or intent of each error generated by LLMs.

First, here is the taxonomy to use:

<taxonomy>
{{ taxonomy }}
</taxonomy>

To complete the task:

1. Carefully read through the entire {{ data_type }}, which
    contains a list of errors.
2. For each error, consult the taxonomy and identify the **single
    most relevant category** that best captures the overall topic
    or intent of that specific error.
3. If no category fits well, use the category 'Other'.
4. Output the result in a JSON format, where each tuple contains
    the error text and its assigned category. Use the following
    format:

{
  "classified_errors": [
    {
      "error_text": "error text 1",
      "category": "category name 1"
    },
```

```
        . . .
      ]
}

5. Do not assign multiple categories to a single error. Choose
    only one that best fits.
That's it! Think carefully and explain your reasoning before
    giving your final category choice for each error.

Assign a single category to each of the following errors:

<{{ data_type }}>
{{ data }}
</{{ data_type }}>

Respond with your categories within json format, one per error. Do
    not include the number, just the category text.
```

## A.2 RESULTED TAXONOMY EXAMPLES

**Category:** Calculation Error

**Label:** Algebraic Simplification Error

**Error Summary:** The model made an algebraic simplification error when combining the substituted terms, resulting in an incorrect evaluation of the expression (output 1 instead of the correct value 2).

**Category:** Unwarranted Assumption

**Label:** Unjustified Geometric Assumption

**Error Summary:** The incorrect response introduced an unjustified geometric assumption (setting the triangle's height equal to half the base, $h = b$) to simplify the equations. This assumption is not derived from the problem conditions and leads to an erroneous computation of the side-ratio, yielding $\frac{\sqrt{2}}{2}$ instead of the correct $\sqrt{2}$.

**Category:** Data Extraction

**Label:** Missing Quantitative Details

**Error Summary:** The response omits all required quantitative details (exact metric values, percentage improvements, and significance markers), providing only vague qualitative statements.

# B  EXPERIMENTAL SETUP

We begin by introducing the conducted experiments, followed by a description of the general configuration shared across them, and conclude with a summary of the compute resources used for each experiment.

We conducted three experiments, which provide examples for the flexible usage of our approach; (1) `ErrorAtlas` Construction (§3, §4): we sample from all selected data and models, (2) Model Comparison: we utilize the `ErrorAtlas` categories and run only stages 1 (Per-Instance Error Analysis) and 3 (Applying the taxonomy) in `ErrorMap` on all predictions of two Gemini models listed in the HELM Capabilities leaderboard. We then present a comparative evaluation between them in Section 5.1. (3) Dataset Taxonomy: We demonstrate the application of `ErrorMap` to generate a taxonomy tailored to a specific dataset, MMLU-Pro benchmark, in Section D.1.

**Failure Threshold**   `ErrorMap` relies on a distinction between failed and successful instances, in non-binary metrics we make this distinction through a threshold. For each benchmark, we rely on a single metric (the primary score in the benchmark if there are multiple), and define for each range of scores what is the threshold considered as error.[5] For datasets evaluated with a binary score, the selection is straightforward. For others, we found that using an approximation of $0.7\%$ of the maximum score per instance yields good results. Further setup details are provided in Appendix B.

**Taxonomy Parameters**   The error categorization had to be well-defined in each of its prompts to provide a specific output. As part of this stage, and following the approach described in Wan et al. (2024), we defined a set of parameters tailored to our case, such as error label batch size, maximum length for category names, and others. The complete list of parameters and their corresponding values is provided in Table A.1.

**Selected Judge**   All experiments were conducted using the `gpt-oss-120b` model (OpenAI, 2025), chosen for its scale and relevance to current state-of-the-art systems. To better leverage its strong reasoning capabilities, we adapted the prompts accordingly. We add maximum 3 ICPs (if any exist) to each prompt.

**Compute**   The required compute for `ErrorMap` depends on the number of incorrect predictions. `ErrorAtlas` creation required approximately 7,200 inferences. Since most of these can run in parallel, the process took approximately 3 hours. The Gemini model comparison required about 2,000 inferences. The MMLU-Pro experiment required approximately 3,500 inferences.

**Reliability validation**   We used the `sentence-transformers/all-MiniLM-L6-v2` model. Changing the embedder did not change results, maybe because our task goes beyond textual similarity and aims to capture the underlying skills implied by the labels.

**Statistical Significance Test**   To assess whether the differences in model distributions are statistically significant, we conducted pairwise comparisons between models. Specifically, we used binomial probability tests to evaluate the likelihood that the observed performance of a weaker model could occur under the distribution of a stronger one.

---

[5] A higher threshold is preferred over a lower one, as it prevents false negatives, ensuring that genuine errors are not mistakenly excluded. False positives, on the other hand, are mostly filtered out during instance-level analysis.

## C    ERRORATLAS DETAILS AND RESULTS

| Dataset Name | Benchmark |
|---|---|
| MMLU-Pro | |
| GPQA | |
| OmniMATH | HELM Capabilities |
| WildBench | |
| IFEval | |
| ACIBench | |
| MedDialog (healthcare magic) | |
| MedDialog (icliniq) | |
| MEDEC | MedHELM |
| MediQA | |
| MedicationQA | |
| MTSamples procedures | |
| MTSamples replicate | |
| QTSumm | |
| NumericNLG | ToRR |
| SciGen | |
| TableBench (data analysis) | |
| HumanEval | |
| HumanEval+ | — |
| MBPP | |
| MBPP+ | |

Table C.1: List of datasets used to create `ErrorAtlas`.

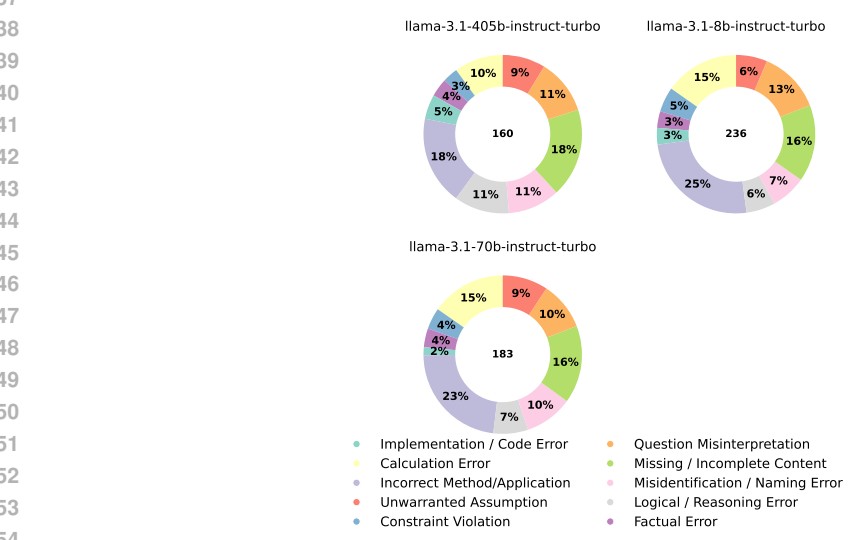

Figure C.1: LLama models distribution on Capabilites benchmark.

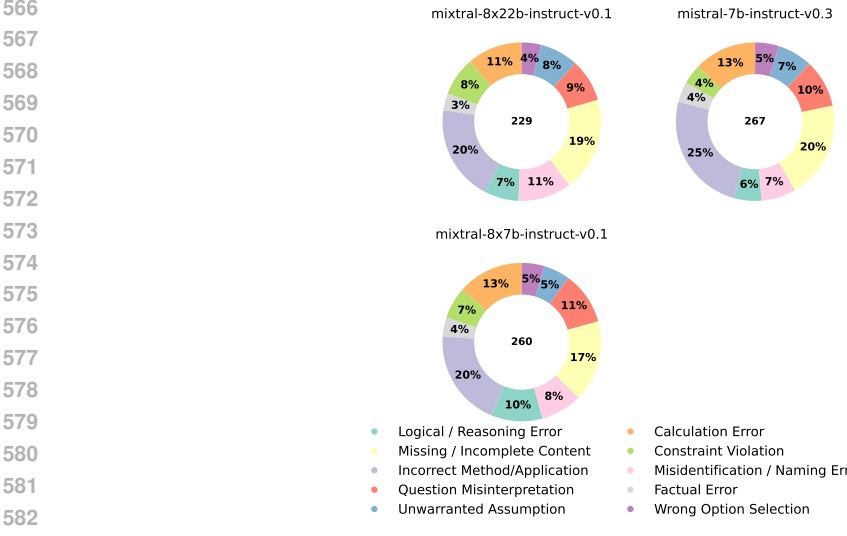

Figure C.2: Mistral AI models distribution on Capabilites benchmark.

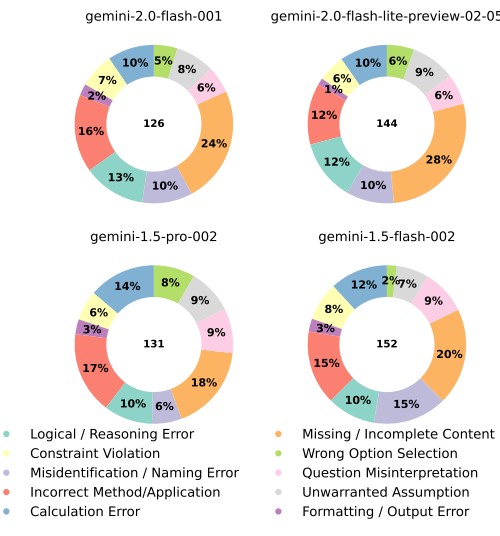

Figure C.3: Gemini models distribution on Capabilites benchmark.

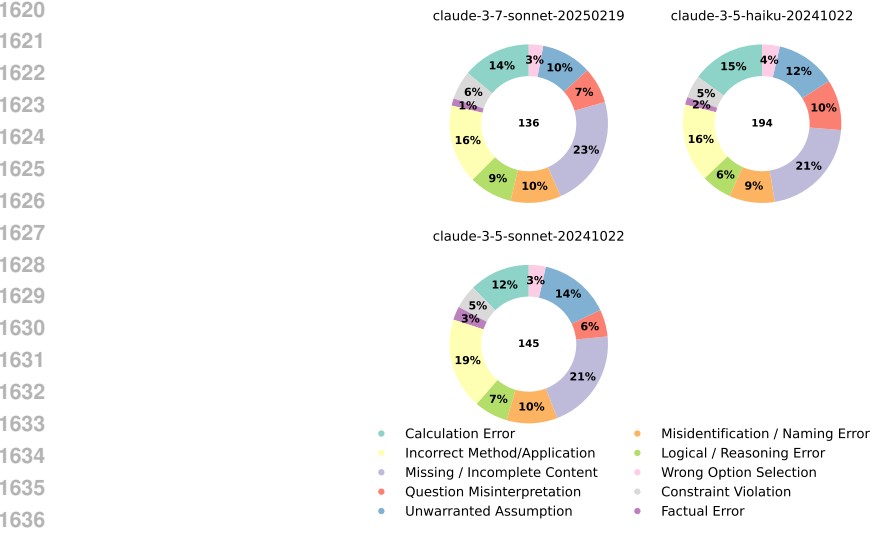

Figure C.4: Claude models distribution on Capabilites benchmark.

| Error Category - Original Result | Error Category - Manually Refined Version |
|---|---|
| Missing / Incomplete Content | Incomplete Content |
| Incorrect Method/Application | Incorrect Method/Application |
| Calculation Error | Calculation Error |
| Question Misinterpretation | Question Misinterpretation |
| Unwarranted Assumption | Unwarranted Assumption |
| Misidentification / Naming Error | Naming Error |
| Constraint Violation | Constraint Violation |
| Logical / Reasoning Error | Reasoning Error |
| Formatting / Output Error | Formatting Error |
| Factual Error | Factual Error |
| Implementation / Code Error | Code Error |
| Irrelevant / Off-Topic | Irrelevant / Off-Topic |
| Language / Grammar Issue | Language Issue |
| Hallucination / Fabricated Content | Hallucination |
| Policy Violation | Policy Violation |
| Refusal / Non-compliant Response | Refusal |
| Repetition / Verbosity | Verbosity |
| Data Extraction / Quantitative Omission | Data Extraction |
| Wrong Option Selection | [Limited informativeness.] |
| Incorrect Table Identification | [This may be resulted because specific table data type, while we aimed for more general categories.] |
| No Response / Empty Output | [Limited informativeness and rare appearance (see C.3).] |
| Plagiarism / Unoriginal Content | [Too rare occurrence (see C.3).] |
| Other | — |
| Hard to Analyze | — |

Table C.2: Comparison between original error categories and their manually refined versions. The Other and Hard to Analyze categories were explicitly added as fallback options. They allow the model to indicate when there is insufficient information to analyze an error or when no other category is a suitable fit. Moreover, 4 categories out of 22 generated by the model were not included in ErrorAtlas and the reasons are mentioned in the table.

| Error Category | Num. of Datasets | Num. of Models | Prevalence (Count) | Prevalence (%) |
|---|---|---|---|---|
| Missing / Incomplete Content | 13 | 73 | 1569 | 22 |
| Incorrect Method/Application | 9 | 60 | 831 | 12 |
| Calculation Error | 8 | 55 | 591 | 8 |
| Question Misinterpretation | 11 | 63 | 462 | 7 |
| Unwarranted Assumption | 8 | 41 | 406 | 6 |
| Misidentification / Naming Error | 10 | 52 | 433 | 6 |
| Constraint Violation | 7 | 60 | 411 | 6 |
| Logical / Reasoning Error | 9 | 50 | 421 | 6 |
| Formatting / Output Error | 8 | 58 | 237 | 3 |
| Factual Error | 10 | 52 | 180 | 3 |
| Implementation / Code Error | 4 | 52 | 176 | 2 |
| Irrelevant / Off-Topic | 11 | 47 | 125 | 2 |
| Wrong Option Selection | 7 | 37 | 170 | 2 |
| Incorrect Table Identification | 3 | 14 | 57 | 1 |
| Language / Grammar Issue | 4 | 48 | 87 | 1 |
| Hard to Analyze | 5 | 29 | 48 | 1 |
| Hallucination / Fabricated Content | 5 | 47 | 99 | 1 |
| No Response / Empty Output | 8 | 23 | 65 | 1 |
| Policy Violation | 8 | 42 | 77 | 1 |
| Refusal / Non-compliant Response | 7 | 25 | 73 | 1 |
| Repetition / Verbosity | 5 | 43 | 83 | 1 |
| Data Extraction / Quantitative Omission | 9 | 40 | 102 | 1 |
| Other | 1 | 1 | 1 | 0 |
| Plagiarism / Unoriginal Content | 1 | 3 | 3 | 0 |

Table C.3: `ErrorAtlas` Statistics: The original categories generated by `gpt-oss-120b`, with the number of models and datasets classified to each category, along with their prevalence in count and percentage.

## C.1 EXAMPLES

### C.1.1 INCOMPLETE RESPONSE EXAMPLES

```
Context:\\
Solve the equation $a^3 + b^3 + c^3 = 2001$ in positive integers.

References:\\
$\boxed{\left[(10,10,1), (10,1,10), (1,10,10)\right]}$

incorrect prediction:
\\ We are asked to solve the equation $a^3 + b^3 + c^3 = 2001$ in
    positive integers.
\\ We can observe that $1^3 = 1$, $2^3 = 8$, $3^3 = 27$, $4^3 = 64
    $, $5^3 = 125$, $6^3 = 216$, $7^3 = 343$, $8^3 = 512$, $9^3 =
    729$, $10^3 = 1000$, $11^3 = 1331$, $12^3 = 1728$, $13^3 =
    2197$.
\\ Since $a, b, c$ are positive integers, we must have $a, b, c \
    le 12$.
\\ Also, we can assume without loss of generality that $a \ge b \
    ge c \ge 1$.
\\ If $a=12$, then $a^3 = 1728$, so $b^3 + c^3 = 2001 - 1728 = 273
    $.
\\ If $b=6$, then $b^3 = 216$, so $c^3 = 273 - 216 = 57$, which is
     not a perfect cube.
\\ If $b=5$, then $b^3 = 125$, so $c^3 = 273 - 125 = 148$, which
    is not a perfect cube.
```

| Error Category | Prevalence (Count) | Prevalence (%) |
|---|---|---|
| Incorrect Method/Application | 600 | 21 |
| Calculation Error | 417 | 14 |
| Missing / Incomplete Content | 345 | 12 |
| Unwarranted Assumption | 321 | 11 |
| Question Misinterpretation | 302 | 10 |
| Misidentification / Naming Error | 280 | 10 |
| Logical / Reasoning Error | 282 | 10 |
| Wrong Option Selection | 136 | 5 |
| Constraint Violation | 56 | 2 |
| Factual Error | 64 | 2 |
| Implementation / Code Error | 2 | 0 |
| Incorrect Table Identification | 1 | 0 |
| Hard to Analyze | 5 | 0 |
| Hallucination / Fabricated Content | 2 | 0 |
| Formatting / Output Error | 6 | 0 |
| No Response / Empty Output | 1 | 0 |
| Policy Violation | 1 | 0 |
| Refusal / Non-compliant Response | 1 | 0 |
| Data Extraction / Quantitative Omission | 14 | 0 |
| Irrelevant / Off-Topic | 5 | 0 |

Table C.4: MMLU-Pro, GPQA and Omni MATH error category statistics.

```
\\ If $b=4$, then $b^3 = 64$, so $c^3 = 273 - 64 = 209$, which is
    not a perfect cube.
\\ If $b=3$, then $b^3 = 27$, so $c^3 = 273 - 27 = 246$, which is
    not a perfect cube.
\\ If $b=2$, then $b^3 = 8$, so $c^3 = 273 - 8 = 265$, which is
    not a perfect cube.
\\ If $b=1$, then $b^3 = 1$, so $c^3 = 273 - 1 = 272$, which is
    not a perfect cube.
\\ If $a=11$, then $a^3 = 1331$, so $b^3 + c^3 = 2001 - 1331 = 670
    $.
\\ If $b=8$, then $b^3 = 512$, so $c^3 = 670 - 512 = 158$, which
    is not a perfect cube.
\\ If $b=7$, then $b^3 = 343$, so $c^3 = 670 - 343 = 327$, which
    is not a perfect cube.
\\ If $b=6$, then $b^3 = 216$, so $c^3 = 670 - 216 = 454$, which
    is not a perfect cube.
\\ If $b=5$, then $b^3 = 125$, so $c^3 = 670 - 125 = 545$, which
    is not a perfect cube.
\\ If $b=4$, then $b^3 = 64$, so $c^3 = 670 - 64 = 606$, which is
    not a perfect cube.
\\ If $b=3$, then $b^3 = 27$, so $c^3 = 670 - 27 = 643$, which is
    not a perfect cube.
\\ If $b=2$, then $b^3 = 8$, so $c^3 = 670 - 8 = 662$, which is
    not a perfect cube.
\\ If $b=1$, then $b^3 = 1$, so $c^3 = 670 - 1 = 669$, which is
    not a perfect cube.
\\ If$a=10$, then $a^3 = 1000$, so $b^3 + c^3 = 2001 - 1000 = 1001
    $.
\\ If$b=10$, then $b^3 = 1000$, so $c^3 = 1001 - 1000 = 1$, so $c
    =1$.
Thus, $(10, 10, 1)$ is a solution.
\\ If$a=1$, $b=10$, $c=10$, then $1^3 + 10^3 + 10^3 = 1 + 1000 +
    1000 = 2001$.
```

```
Thus, $(1, 10, 10)$ is a solution.
\\ If$a=10$, $b=1$, $c=10$, then $10^3 + 1^3 + 10^3 = 1000 + 1 +
    1000 = 2001$.
Thus, $(10, 1, 10)$ is a solution.
\\ If $a=10$, $b=10$, $c=1$, then $10^3 + 10^3 + 1^3 = 1000 + 1000
    + 1 = 2001$.
\\ Thus, $(10, 10, 1)$ is a solution.

Final Answer:
\\ The final answer is $\boxed{(1, 10, 10)}$
%
```

Listing 1: Sample JSON Output

```
{
  "required_criteria": [
    {
      "criterion": "Establish an upper bound for the variables (show that
          a, b, c $\leq$ 12).",
      "present_in_wrong": true,
      "quality": "correct",
      "evidence": "Since $a, b, c$ are positive integers, we must have $a
          , b, c \\le 12$.",
      "comment": "The bound is correctly derived from $13^3 > 2001$."
    },
    {
      "criterion": "Assume without loss of generality an ordering (e.g.,
          a $\geq$ b $\geq$ c) to reduce duplicate work.",
      "present_in_wrong": true,
      "quality": "correct",
      "evidence": "Also, we can assume without loss of generality that $a
          \\ge b \\ge c \\ge 1$.",
      "comment": "Proper ordering is stated."
    },
    {
      "criterion": "Systematically examine all feasible values of a (
          starting from the maximum) and for each check whether the
          remaining sum can be expressed as two cubes.",
      "present_in_wrong": true,
      "quality": "partially correct",
      "evidence": "The response checks a $\eq$ 12, 11, 10 but stops there
          ; it does not examine a $\eq$ 9, 8, $\ldots$ down to the lower
          bound.",
      "comment": "Missing cases means the search is incomplete; the
          criterion is only partially satisfied."
    },
    {
      "criterion": "Identify the concrete solution(s) when a = 10 (i.e.,
          b = 10, c = 1).",
      "present_in_wrong": true,
      "quality": "correct",
      "evidence": "If $a=10$, then $b^3 + c^3 = 1001$ $\ldots$ $b=10$, $c
          =1$ $\ldots$ Thus, $(10,10,1)$ is a solution.",
      "comment": "Correctly finds the core solution."
    },
    {
      "criterion": "Account for all permutations of the found unordered
          triple to list every ordered solution.",
      "present_in_wrong": true,
      "quality": "partially correct",
      "evidence": "The answer mentions $(1,10,10)$ and $(10,1,10)$ but
          the final boxed answer only shows $(1,10,10)`. ",
```

```
37        "comment": "Permutations are recognized, yet the final response
                lists only one of the three ordered triples."
38      },
39      {
40        "criterion": "Conclude that the listed permutations are the only
                positive-integer solutions to
41        $a^3+b^3+c^3=2001$.",
42        "present_in_wrong": false,
43        "quality": "incorrect",
44        "evidence": "",
45        "comment": "The response never explicitly states that no other
                solutions exist."
46      }
47    ],
48    "final_answer": {
49      "error_summary": "The answer fails to present the complete set of
            solutions and does not demonstrate that no other triples exist;
            it only outputs one ordered triple while ignoring the other two
            permutations and omits a final uniqueness argument.",
50      "error_title": "Incomplete solution set"
51    }
52 }
53
54 Example_id: id3293
55 Model: google/gemini-2.0-flash-lite-preview-02-05
56 Dataset: omni_math
```

Context:
what does tasmar contain

References:
ACTIVE INGREDIENT/ACTIVE MOIETY
Tolcapone (UNII: CIF6334OLY)

INACTIVE INGREDIENTS
Lactose Monohydrate (UNII: EWQ57Q8I5X)
povidone K30 (UNII: U725QWY32X)
sodium starch glycolate type a potato (UNII: 5856J3G2A2)
talc (UNII: 7SEV7J4R1U)
magnesium stearate (UNII: 70097M6I30)
titanium dioxide (UNII: 15FIX9V2JP)
triacetin (UNII: XHX3C3X673)
sodium lauryl sulfate (UNII: 368GB5141J)
ferric oxide yellow (UNII: EX438O2MRT)
ferric oxide red (UNII: 1K09F3G675)
MICROCRYSTALLINE CELLULOSE (UNII: OP1R32D61U)
ANHYDROUS DIBASIC CALCIUM PHOSPHATE (UNII: L11K75P92J)
ETHYLCELLULOSE, UNSPECIFIED (UNII: 7Z8S9VYZ4B)

incorrect prediction:
Tasmar is a brand name for the drug tolcapone. Tolcapone is a
    medication used in the treatment of Parkinson's disease. It
    works as a catechol–O–methyltransferase (COMT) inhibitor,
    which helps to increase the levels and duration of action of
    levodopa, a common medication used to manage Parkinson's
    symptoms. Tasmar is typically used in combination with
    levodopa/carbidopa therapy to improve its effectiveness. It is
    important to note that Tasmar can have serious side effects,
    including liver toxicity, so its use is carefully monitored by
    healthcare providers.

Listing 2: Sample JSON Output

```json
{
  "required_criteria": [
    {
      "criterion": "State the active ingredient of Tasmar",
      "present_in_wrong": true,
      "quality": "correct",
      "evidence": "Tasmar is a brand name for the drug tolcapone.",
      "comment": "The response correctly identifies Tolcapone as the
          active ingredient."
    },
    {
      "criterion": "List all inactive ingredients of Tasmar as given in
          the reference",
      "present_in_wrong": false,
      "quality": "incorrect",
      "evidence": "",
      "comment": "The response does not provide any of the inactive
          ingredients; it instead discusses clinical use and safety."
    }
  ],
  "final_answer": {
    "error_summary": "The model focused on the pharmacological
        description of Tasmar rather than enumerating its ingredient
        composition, omitting the required list of inactive ingredients."
    ,
    "error_title": "Omission of Required Ingredient List"
  }
}

example_id: id553
model: openai/gpt-4o-2024-05-13
dataset: medhelm_v2_medication_qa
```

### C.1.2 QUESTION MISINTERPRETATION EXAMPLES

```
Context:
This question refers to the following information.
"To slacken the tempo would mean falling behind. And those who
    fall behind get beaten. But we do not want to be beaten. No,
    we refuse to be beaten! One feature of the history of old
    Russia was the continual beatings she suffered because of her
    backwardness. She was beaten by the Mongol khans. She was
    beaten by the Turkish beys. She was beaten by the Swedish
    feudal lords. She was beaten by the Polish and Lithuanian
    gentry. She was beaten by the British and French capitalists.
    She was beaten by the Japanese barons.
All beat her--because of her backwardness,
because of her military backwardness, cultural
backwardness, political backwardness, industrial
backwardness, agricultural backwardness. They beat her
because it was profitable and could be done with impunity.
You remember the words of the pre-revolutionary poet: "You are
    poor and abundant, mighty and impotent, Mother Russia."
Those gentlemen were quite familiar with the verses of the old
    poet. They beat her, saying: "You are abundant," so one can
    enrich oneself at your expense. They beat her, saying: "You
    are poor and impotent,"
so you can be beaten and plundered with impunity. Such is the law
    of the exploiters--to beat the backward and the weak. It is
    the jungle law of capitalism.
```

You are backward, you are weak--therefore you are wrong;
hence you can be beaten and enslaved. You are mighty--
therefore you are right; hence we must be wary of you.
That is why we must no longer lag behind."
Joseph Stalin, speech delivered at the first All-Union Conference
    of Leading Personnel of Socialist Industry, February 4, 1931
The reference to the "continual beatings" most directly supports
    which of the following decisions in later Soviet foreign
    policy?

Candidate Answers:
A) The declaration of war against Japan at the end of World War II
B) The blockade of Berlin
C) The support for Cuban Revolution
D) The decision to build the Berlin Wall
E) The imposition of the Brezhnev Doctrine against anti-Soviet
    protesters in Europe
F) The signing of the Molotov-Ribbentrop Pact
G) The launch of Sputnik
H) The diplomatic recognition of the People's Republic of China
I) The invasion of Afghanistan
J) The creation of the Warsaw Pact

References:
J

incorrect prediction:
The passage emphasizes the historical vulnerability and
    backwardness of Russia, which led to exploitation and
    aggression by foreign powers. Stalin's speech highlights the
    need to avoid lagging behind in industrial and military
    strength to prevent future exploitation. This sentiment aligns
     with the Soviet Union's decision to assert its power and
    influence to avoid being perceived as weak or backward.

Let's evaluate the options:

- **A. The declaration of war against Japan at the end of World
    War II:** This decision was driven by Soviet strategic
    interests and the desire to regain territory, but it is not
    directly tied to the theme of avoiding backwardness.
- **B. The blockade of Berlin:** This was a Cold War strategy to
    assert Soviet power in Europe, but it does not directly relate
     to the historical theme of backwardness.
- **C. The support for Cuban Revolution:** Supporting revolutions
    abroad was part of Soviet efforts to expand influence, but it
    is not directly tied to the theme of backwardness.
- **D. The decision to build the Berlin Wall:** This was a measure
     to prevent East Germans from fleeing to the West, but it is
    not directly tied to the theme of backwardness.
- **E. The imposition of the Brezhnev Doctrine against anti-Soviet
     protesters in Europe:** This was a policy to maintain Soviet
    control over Eastern Europe, but it does not directly address
    the theme of backwardness.
- **F. The signing of the Molotov-Ribbentrop Pact:** This non-
    aggression pact with Nazi Germany was a pragmatic decision to
    buy time and avoid conflict, but it does not directly relate
    to the theme of backwardness.

```
- **G. The launch of Sputnik:** This was a demonstration of Soviet
    technological advancement and strength, directly addressing
    the need to avoid being perceived as backward.
- **H. The diplomatic recognition of the People's Republic of
    China:** This was a geopolitical move, but it does not
    directly relate to the theme of backwardness.
- **I. The invasion of Afghanistan:** This was an attempt to
    expand Soviet influence, but it does not directly address the
    theme of backwardness.
- **J. The creation of the Warsaw Pact:** This was a military
    alliance to counter NATO, asserting Soviet strength, but it is
    not directly tied to the theme of backwardness.

The **launch of Sputnik** (G) best aligns with the theme of Stalin
    's speech, as it demonstrated Soviet technological and
    industrial progress, countering the historical narrative of
    backwardness and vulnerability.

The correct answer is (G).
```

Listing 3: Sample JSON Output

```json
1  {
2    "required_criteria": [
3      {
4        "criterion": "Recognize that the passage stresses historical
              exploitation of Russia because of its backwardness and the need
               to avoid lagging behind militarily.",
5        "present_in_wrong": true,
6        "quality": "partially correct",
7        "evidence": "The model notes: \"The passage emphasizes the
              historical vulnerability and backwardness of Russia, which led
              to exploitation and aggression by foreign powers.\"",
8        "comment": "The model correctly identifies the theme of
              backwardness, but does not link it specifically to a security-
              oriented foreign-policy response."
9      },
10     {
11       "criterion": "Link the theme of avoiding backwardness to a foreign-
              policy decision that provides collective security against
              external powers.",
12       "present_in_wrong": true,
13       "quality": "incorrect",
14       "evidence": "The model argues that the launch of Sputnik \"directly
               addresses the need to avoid being perceived as backward.\"",
15       "comment": "Sputnik is a technological showcase, not a collective
              security measure; the passage calls for defending against
              exploitation, not a prestige project."
16     },
17     {
18       "criterion": "Exclude options that pertain to internal achievements
               or isolated diplomatic moves that do not create a defensive
              alliance.",
19       "present_in_wrong": true,
20       "quality": "incorrect",
21       "evidence": "The model dismisses many options (e.g., Berlin Wall,
              Brezhnev Doctrine) as ``not directly tied to the theme of
              backwardness,'' yet incorrectly selects Sputnik, also an
              internal achievement.",
22       "comment": "The exclusion reasoning is inconsistent; the correct
              exclusion should keep the Warsaw Pact and reject purely
              symbolic actions."
23     },
```

```
24        {
25          "criterion": "Select the Warsaw Pact (option J) as the decision
                 that most directly embodies the passage's call for collective
                 defense against exploitation.",
26          "present_in_wrong": false,
27          "quality": "incorrect",
28          "evidence": "The model never mentions the Warsaw Pact and instead
                 chooses option G (Sputnik).",
29          "comment": "Failure to consider the Warsaw Pact indicates the model
                 missed the central security implication of the passage."
30        }
31      ],
32      "final_answer": {
33        "error_summary": "The model misinterpreted the passage's emphasis on
                 historical exploitation and the need for collective military
                 security, treating it as a call for technological prestige and
                 consequently selecting Sputnik instead of the Warsaw Pact.",
34        "error_title": "Misinterpretation of Historical Context Leading to
                 Wrong Policy Choice"
35      }
36 }
37 model: deepseek-ai/deepseek-v3
38 dataset: mmlu_pro_old
39 example_id: id5031
```

# D    ERRORMAP APPLICABILITY

## D.1    BENCHMARK STAKEHOLDERS

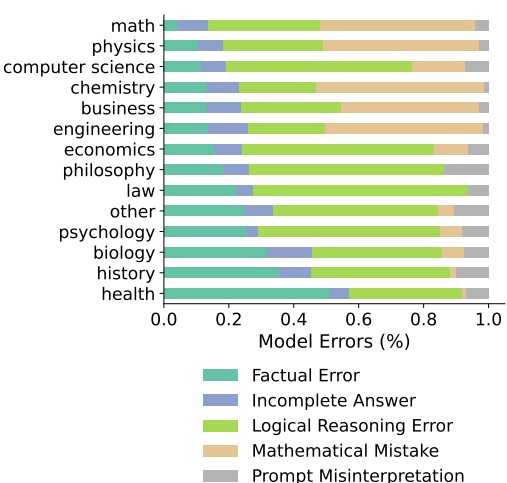

Figure D.1: Differences in error category distributions across domain categories in MMLU-Pro.

# E    ERRORMAP EVALUATION

| Stage | Experiment | Accuracy (%) |
|---|---|---|
| Per-Instance | ErrorAtlas | .89 |
| | Gemini comparison | .90 |
| | MMLU-Pro taxonomy | .94 |
| Taxonomy | ErrorAtlas | .92 |
| | Gemini comparison | .93 |
| | MMLU-Pro taxonomy | .91 |

Table E.1: `ErrorMap` and `ErrorAtlas` Evaluation results.

| Original Results | Results after Prompt Variation 1 | Results after Prompt Variation 2 |
|---|---|---|
| Calculation Error | Calculation Error | Incorrect Computation or Derivation |
| Logical / Reasoning Error | Logical Reasoning Error | Misapplication of Concept or Method |
| Missing / Incomplete Content | Missing Required Information | Task Requirement Violation |
| Constraint Violation | Formatting Violation | Formatting and Structural Violations |
| No Response / Empty Output | Incomplete Solution | Length Constraint Violation |
| Language / Grammar Issue | Incorrect Tone / Style | Improper Language or Style |
| Data Extraction / Quantitative Omission | Data Extraction Error | Invalid Assumptions or Input Conditions |
| Incorrect Method/Application | Misapplied Principle/Theorem | Misapplication of Concept or Method |
| Factual Error | Factual Inaccuracy | Factual Inaccuracy |
| Incorrect Table Identification | Incorrect Model Assumption | Invalid Assumptions or Input Conditions |
| Formatting / Output Error | Formatting Violation | Formatting and Structural Violations |
| Question Misinterpretation | Misinterpretation of Prompt | Misinterpretation of Prompt or Data |
| Implementation / Code Error | Code / Implementation Omission | Task Requirement Violation |
| Unwarranted Assumption | Incorrect Model Assumption | Invalid Assumptions or Input Conditions |
| Irrelevant / Off-Topic | Irrelevant Content Inclusion | Irrelevant or Extraneous Content |
| Wrong Option Selection | Wrong Answer Choice Selection | Incorrect Answer Selection/Mapping |
| Hard to Analyze | Hard to Analyze | Hard to Analyze |
| Repetition / Verbosity | Insufficient Depth / Analysis | Length Constraint Violation |
| Policy Violation | Policy Violation | Policy Violation (Refusal/Disallowed Content) |
| Refusal / Non-compliant Response | Refusal Error | Policy Violation (Refusal/Disallowed Content) |
| Hallucination / Fabricated Content | Hallucinated Information | Factual Inaccuracy |
| Misidentification / Naming Error | — | — |
| Plagiarism / Unoriginal Content | — | — |
| Domain Knowledge Error | — | — |
| Algebraic / Geometric Manipulation Error | Calculation Error | Incorrect Computation or Derivation |
| Probability / Statistical Error | Calculation Error | Incorrect Computation or Derivation |
| Other | Other | Other |

Table E.2: Comparative analysis of `ErrorAtlas` final error categories across prompt variants and sampling ratio and seeds. Variation 1 was created with 5% random sample of the data, while Variation 2 was created with 15%.

## F    LIMITATIONS

**Predictions Signal**    While our method relies on model outputs, we acknowledge that the prediction signal can be partial compared to what actually happens inside the model (as opposed to white-box interpretability).

**Informative Prediction Dependence**    `ErrorMap` focuses on predictions as the primary basis for analysis. This approach technically depends on informative predictions. If a model cannot be run in generative mode and does not explain its response (CoT is also acceptable), then its responses cannot be analyzed.

**Error Category**    We acknowledge that error categories are inherently soft, that is, a single mistake may reflect multiple underlying issues. For example, a model incorrectly stating that "mRNA carries amino acids to the ribosome" could indicate either a factual error or confusion about molecular roles.

**ErrorAtlas Generality**    While we have tried to create `ErrorAtlas` in the most varied way possible, there may be cases where it does not represent certain specific domains well.

**LLM-based technique**  This work makes use of LLMs to analyze mistakes made by LLMs themselves. While this approach is somewhat circular, verification and comparison are generally easier than generation (Simonds et al., 2025; Pang et al., 2023; Lin et al., 2024). However, this assumption may not always hold in practice.

## G   USAGE IN AI

In this work, we used AI models exclusively for language-related tasks, such as rephrasing and surface-level linguistic transformations. It was further used for minor improvements to code style across the repo.

