# OpenReview forum: "ErrorMap and ErrorAtlas: Charting the Failure Landscape of Large Language Models"
_ICLR.cc/2026/Conference — Submitted to ICLR 2026_

### Official Review · Reviewer_7RK4 · 2025-10-25

**Soundness:** 2
**Presentation:** 2
**Contribution:** 2
**Rating:** 2
**Confidence:** 3

**Summary:**

The article has adjusted the default Latex template, significantly reducing the margins. This allows for more content to be written within the required number of pages. I believe this is unfair to other paper authors.

**Strengths:**

N/A

**Weaknesses:**

N/A

**Questions:**

N/A

---

> ### Author Response · Authors · 2025-11-19
>
> We thank the reviewer for pointing this out. We acknowledge that the margin adjustment was an unintended mistake on our part. Our intention was not to gain any unfair advantage, and we sincerely apologize for this oversight.
>
> We updated the paper version to fit the required format and we would greatly appreciate a review regarding the paper’s content, as your insights on the technical contribution would help us improve the work further.

---

### Official Review · Reviewer_h4hL · 2025-10-31

**Soundness:** 2
**Presentation:** 2
**Contribution:** 2
**Rating:** 4
**Confidence:** 5

**Summary:**

The paper introduces error map, which is a method to systematically analyze why LLMs fail rather than just where they fail. And for that, they use LLM-based analysis in three stages to create error taxonomies.

The pipeline is three-stage and it turns model mistakes on benchmark into layered error taxonomy. And the error atlas is a static taxonomy distilled by applying error map across many models and datasets.

This atlas is built by sampling failures from 21 datasets in many models to produce high-level categories like factual error, misinterpretation, incomplete content, etc.

**Strengths:**

1. The paper introduces a clear general pipeline for systematic error analysis.

2. The error atlas spans about 21 datasets and many models which enables cross-model and cross-benchmark comparisons.

3. There is a concrete taxonomy with readable category names and definitions which is useful.

**Weaknesses:**

The usage of LLMs to judge LLMs failure modes is fundamentally problematic. And while the authors acknowledge this, it does introduce systematic blind spots with the judge model shares failure modes with evaluated models.

The validation is not extremely strong. The 92% taxonomy accuracy comes from the same LLM judge and only 53% similarity occurs across prompt variations which suggest some instability. If the authors had done some Newman evaluation and had published agreement scores between the two evaluations, that would have increased the trust.

Only a 10% sampling rate seems somewhat arbitrary without power analysis and the binomial test could be made more robust as it is currently quite simplistic.

The single first-order attribution oversimplifies the cascading failures and the small error threshold is quite data-set-agnostic and unjustified. Also, the use of a single-judge model, gpt-oss, introduces specific biases.

Forcing errors into single categories also loses nuance. Multi-level classification would probably be more appropriate.

There is a minor inconsistency in 73 models being said in the abstract, whereas later text saying 75 models.

The authors pruned categories that appear in fewer than 20% of the datasets which could draw up informative but niche failure modes.

**Questions:**

Can you please clarify whether the total number of models is 73 or 75, And ensure consistency across sections and figures.

You sample about 10% of failures per model dataset pair. How stable are the resulting categories and proportions if the sampling rate is varied or if you stratify by error type rarity or task family?

When pruning categories that appear in less than 20% of the datasets, you may delete real safety critical or domain-specific failure modes. Can you provide a solution to that?

The robustness is tested via prompt variations and shows a decent amount of moderate similarity. Can you please do cross-judge robustness, for instance, comparing different LLMs in stage 1 and stage 3?

What happens at category boundaries? Several categories may overlap in practice. For instance, incomplete content vs formatting error or reasoning error vs incorrect method. Can you quantify these intercategory confusions? What are the priority rules when multiple categories fit?

---

> ### Author Response · Authors · 2025-11-19
>
> We thank the reviewer for their valuable feedback and appreciate the acknowledgement of our clear and useful pipeline.
>
> 1. While using LLMs as judges may introduce potential blind spots, we emphasize that the judge’s task is fundamentally different from that of the model and is considerably easier. This is because judging primarily involves comparison and verification rather than generation, which are generally recognized as easier tasks ([1], [2], [3]).
>
> 2.  While the BERT scores were moderate, we clarified in lines 422–427 that a manual analysis explains these results. Furthermore, we emphasize that the primary contribution lies in the final list of categories. As demonstrated in **Section 6.2** and **Table E.2**, these variations do not impact the overall outcome. Specifically, we compare the resulting categories across different runs and show that the list of categories remains almost the same.
>
> 3. Our goal was to avoid running all examples while ensuring representativeness. As shown in **Table E.2**, sampling at 5% and 15% does not change the final results.
>    Could you clarify what you mean by *“the binomial test could be made more robust”*?
>
> 4. We focused on the first major error to capture the most informative signal about what the model struggles with. If you have alternative approaches to address this, we would greatly appreciate your suggestions.
>
> 5. We acknowledge that any single model can introduce bias. We would welcome recommendations on efficient strategies to mitigate this further.
>
> 6. Although forcing categories has limitations, allowing multiple categories also introduces drawbacks. For example, an error might be 70% category X and 30% category Y, and counting it for both may misrepresent the true error distribution. Moreover, an error that fits three or four categories can bias the results when each category is counted toward frequency. Hence, we adopted a single-category assignment as an approximation.
>
> 7. Thank you for noting the inconsistency in the number of models; the correct number is 73, and we will ensure this is consistent throughout the text.
>
> 8. We present both the original (Table C.3) and pruned results (Table 1) to demonstrate that no informative points are lost, only data-biased ones are removed.
>
> ---
>
> ### Responses to Your Questions:
> 9. We referred to the models above.
> 10. The effect of sampling does not appear to significantly influence category distributions or proportions. However, task selection has greater potential impact, as different tasks introduce varying levels of difficulty for the models. For this reason, when creating the Atlas, we aimed to keep the task set as diverse as possible.
> 11. We have explained the pruning process above.
> 12. A cross-judge robustness check could further strengthen our results, and we will consider adding this in future work.
> 13. While we report category frequencies, we view the list of categories as the primary result, as it highlights the main failure modes of the models. Our goal is not to provide precise frequency estimates for each category, which is challenging to do reliably. Instead, we focused on achieving the best possible categorization fit, aiming to provide deeper insights that can inform future research on LLMs.
>
> Thank you again for investing your time in the review! If we have addressed your concerns, we would greatly appreciate it if you could update our score accordingly.
>
> ---
>
> **References**
> [1] Simonds, Toby et al. *RLSR: Reinforcement Learning from Self Reward.* (2025).
> [2] Pang, Jing-Cheng et al. *Language Model Self-improvement by Reinforcement Learning Contemplation.* ArXiv abs/2305.14483 (2023): n. pag.
> [3] Lin, Bill Yuchen et al. *WildBench: Benchmarking LLMs with Challenging Tasks from Real Users in the Wild.* ArXiv abs/2406.04770 (2024): n. pag.

---

### Official Review · Reviewer_ECKn · 2025-11-02

**Soundness:** 3
**Presentation:** 3
**Contribution:** 2
**Rating:** 6
**Confidence:** 3

**Summary:**

Traditional benchmark evaluations can indicate when a model fails but offer little insight into why it fails.To address this limitation, the authors propose a new diagnostic framework called ErrorMap, and, based on it, construct a static taxonomy of model errors named ErrorAtlas.They further analyze the error distributions across different models and demonstrate how this approach can be applied in multiple scenarios, including model diagnosis, benchmark design, and domain-specific evaluation. Additionally, the paper presents experimental validation of ErrorMap’s reliability, coverage, accuracy, and robustness.

**Strengths:**

- This work approaches LLM evaluation from a new perspective—explaining why models fail rather than merely identifying when they fail. By constructing a systematic error analysis framework, the authors contribute a method that is both theoretically meaningful and practically valuable. It helps model developers identify weaknesses, reveal capability gaps, and provides a scientific foundation for model improvement and evaluation.
- The core framework, ErrorMap, is conceptually clear and logically structured. Its derivative system, ErrorAtlas, provides a concrete and reusable static taxonomy for analyzing model failures. The paper also includes numerous case studies and quantitative results, enabling readers to clearly understand the method’s effectiveness in practice.

**Weaknesses:**

- The paper’s main contribution is a tool-based framework for analyzing LLM errors, whose greatest value lies in broad adoption. However, the authors have not yet released the code , making it impossible to evaluate the system’s real-world performance, stability, and efficiency.In my opinion,this limitation significantly weakens the practical impact and overall value of the work.
- The per-instance diagnostic stage of ErrorMap relies heavily on an LLM-as-judge mechanism. Experimental results show that label consistency across prompt variations is only 53% (measured by Sentence-BERT similarity), indicating that the process is quite sensitive to prompt design.For a tool intended for practical use, stability is crucial.If simply modifying a prompt or substituting a different judge model could alter the core conclusions of the ErrorMap framework, this would undoubtedly weaken its reliability and reference value.And although the paper briefly discusses this issue, it lacks a systematic analysis of its potential impact.

**Questions:**

N/A

---

> ### Author Response · Authors · 2025-11-19
>
> We thank the reviewer for their valuable feedback and appreciate the recognition that our approach is both theoretically meaningful and practically relevant.
>
> Regarding your concern about code availability. Could you have possibly missed it? As stated in the paper, we will release both the code and the dataset upon acceptance and shared an anonymous code version in the paper (https://anonymous.4open.science/r/ErrorMap-BDBC). Thus, we believe that the fact that the code is “not yet released” should not be an issue for evaluation.
>
> Concerning the per-instance stage: while the BERT scores were moderate, we clarified in lines 422–427 that a manual analysis explains these results. Furthermore, we emphasize that the primary contribution lies in the final list of categories. As demonstrated in Section 6.2 and Table E.2, these variations do not impact the overall outcome. Specifically, we compare the resulting categories across different runs and show that the list of categories remains almost the same.
>
> If we have addressed your concerns, we would greatly appreciate it if you could update our score accordingly.

---

### Meta-Review · Area_Chair_ZiKL · 2026-01-06

**Summary:**

This paper argues that benchmark accuracy alone obscures why LLMs fail, and proposes ErrorMap, a general diagnostic pipeline that uses LLM-based analysis to assign each failure to an underlying failure cause. By applying ErrorMap at scale across many model–dataset pairs, the authors derive ErrorAtlas, a taxonomy intended to capture recurring error types and enable: 1, model “failure signature,” 2 benchmark diagnosis (what datasets measure in practice), and 3, more targeted model improvement.

Across reviewers, there is agreement that the problem framing is important and the proposed pipeline is conceptually clear. Major concerns still exist in whether the approach is sufficiently reliable and reproducible given its reliance on LLM-as-judge at multiple stages, and whether the presented validation is strong enough to support the paper’s intended practical use

Strengths
1. Shifting evaluation from “where models fail” to “why they fail” is useful for both model developers and benchmark designers
2. The atlas is built across many datasets and models, enabling comparisons across tasks and model families

Weaknesses
1, The most significant concern is that using LLMs to judge LLM failures can introduce systematic bias—especially if the judge shares failure modes with evaluated models. While the authors argue judging is “easier than generation,” reviewers request stronger evidence that the judge is not imposing its own bias on the taxonomy.
2. The reported ~53% consistency across prompt variants raises doubt about reliability for per-instance diagnosis.
3. The “first major error” design and single-category assignment may not capture cascading or multi-label errors.

**Reviewer Concerns:**

Addressed

1. Code availability
2. Authors state that sampling at 5% and 15% yields similar outcomes, suggesting sampling rate is not the dominant factor.
3. Authors acknowledge the margin adjustment as an unintended mistake.

Still outstanding / not fully resolved

see weakness

**Reviewer Scores:**

ECKn: likely to stay at 6
h4hL: likely to stay at 4, the main objections (judge bias, circular validation, pruning, single-label) remain largely unaddressed empirically.
7RK4: could increase from 2 to 3 modestly if the format issue is corrected, but since they did not assess technical content, the final impact on their score is uncertain

---

### Decision · Program_Chairs · 2026-01-26

Reject